# A Multitask CNN for Near-Infrared Probe: Enhanced Real-Time Breast Cancer Imaging

**DOI:** 10.3390/s25082349

**Published:** 2025-04-08

**Authors:** Maryam Momtahen, Farid Golnaraghi

**Affiliations:** School of Mechatronic Systems Engineering, Simon Fraser University, Surrey, BC V3T 0A3, Canada; maryam_momtahen@sfu.ca

**Keywords:** deep learning, convolutional neural networks, NIRscan, breast cancer imaging, data augmentation, image reconstruction, classification, tumor localization

## Abstract

The early detection of breast cancer, particularly in dense breast tissues, faces significant challenges with traditional imaging techniques such as mammography. This study utilizes a Near-infrared Scan (NIRscan) probe and an advanced convolutional neural network (CNN) model to enhance tumor localization accuracy and efficiency. CNN processed data from 133 breast phantoms into 266 samples using data augmentation techniques, such as mirroring. The model significantly improved image reconstruction, achieving an RMSE of 0.0624, MAE of 0.0360, R^2^ of 0.9704, and Fuzzy Jaccard Index of 0.9121. Subsequently, we introduced a multitask CNN that reconstructs images and classifies them based on depth, length, and health status, further enhancing its diagnostic capabilities. This multitasking approach leverages the robust feature extraction capabilities of CNNs to perform complex tasks simultaneously, thereby improving the model’s efficiency and accuracy. It achieved exemplary classification accuracies in depth (100%), length (92.86%), and health status, with a perfect F1 Score. These results highlight the promise of NIRscan technology, in combination with a multitask CNN model, as a supportive tool for improving real-time breast cancer screening and diagnostic workflows.

## 1. Introduction

Breast cancer is the most common cancer among women and the second leading cause of cancer-related deaths worldwide [1]. The World Health Organization (WHO) estimates that approximately 2.1 million women are diagnosed annually with breast cancer [2]. Early detection significantly improves survival rates, making it a critical aspect of breast cancer management [1]. While advancements in imaging technologies have contributed to reducing breast cancer mortality [3,4], challenges remain, especially in dense breast tissues. Despite survival rates of 89% and 82% for 5-year and 10-year survival in Canada, breast cancer continues to be the leading cause of cancer-related death among women [5].

Breast cancer detection generally begins with a Clinical Breast Exam (CBE), where abnormalities are identified through physical palpation. However, small tumors, especially those under 2 cm, may be missed, leading to false negatives [6]. Subsequent imaging tests are typically recommended, including mammography, Magnetic Resonance Imaging (MRI), and ultrasound (US). Mammography, while effective in detecting tumors, has limitations, particularly for younger women and those with dense breast tissue, where it shows lower sensitivity and specificity [7]. Although more sensitive for detecting tumors in dense tissue, MRI is costly and requires specialized staff [8,9]. Ultrasound, though radiation-free and safe, suffers from low specificity. Following imaging, a biopsy is often recommended to confirm the presence of cancer, though it is invasive and costly and requires specialized equipment [10]. Despite its diagnostic accuracy, biopsies can cause anxiety and necessitate additional follow-up tests, further complicating the diagnostic process.

Current imaging techniques struggle with sensitivity and specificity, especially in dense breast tissues, often resulting in delayed or inaccurate diagnoses. Given these challenges, there is a need for improved diagnostic tools that can detect early-stage breast cancer more reliably and efficiently [1].

This paper is part of an ongoing research program aimed at developing a proprietary Near-infrared Scan (NIRscan) probe (US Patent: US11457818B2) for early breast cancer detection. The NIRscan probe, designed to detect cancerous tissue by assessing structural and functional features of biological tissue through diffraction differences in the near-infrared spectrum, offers a promising solution. Previous generations of the NIRscan probe were tested on optical phantoms and breast tissues, and the optical tissue properties were successfully established. The 2016 version of the probe was tested in clinical trials at the Jim Pattison Outpatient Care and Surgery Centre (JPOCSC) with known breast cancer patients, showing 95% accuracy (sensitivity: 0.947, specificity: 0.842) [11]. However, it had limitations, including the inability to perform real-time tumor localization and the production of lab-generated images that required manual adjustments due to artifacts, leading to extended scanning times.

To overcome these challenges, another study focused on improving the NIRscan probe’s capabilities. Enhancements included upgrades to the linear charge-coupled device (CCD) sensor, light intensity control, noise reduction, and the data collection process. However, the previous imaging algorithm, which relied on the diffusion equation (DE), proved inadequate for the new version due to non-collimated light sources. In another study, we developed a modified diffusion equation (MDE) for the new version of the probe, enabling the determination of optical properties and differentiation between normal and abnormal breast phantoms. However, this method was time-consuming.

To address the MDE limitations in [11], we leveraged the application of machine learning algorithms to generate optical properties and images. In our recent study published in Sensors [12], we employed regression analysis with k-fold cross-validation to train a model on phantom data to predict optical properties in healthy and unhealthy breast tissues from 15 breast cancer patients. While the model showed high accuracy, it was unable to generate 3D images directly from optical properties.

In [11], we introduced a new analytical imaging method to calculate absorption coefficients (μa) and map them onto images for optical phantoms and patients. While the method could successfully generate images, the process required manual intervention, motivating the development of an automated method for real-time feedback.

This study introduces a customized convolutional neural network (CNN) model for the NIRscan probe, designed to automate breast cancer image prediction using direct reflectance data, thereby enabling the generation of automated real-time images without the need for derived optical properties like absorption coefficients. To train the CNN model, we create a dataset consisting of 128 reflectance values for LED1 and 128 values for LED2 for each sample collected by the probe, along with corresponding images generated by our analytical imaging technique. The dataset includes 133 samples from breast phantoms. We expanded this dataset to 266 samples using mirroring techniques to enhance variability.

The CNN model was trained to reconstruct accurate, real-time images from unseen data, demonstrating significant potential for clinical use in real-time breast cancer detection. While the model achieved high accuracy in image reconstruction, it exhibited limitations in precisely predicting depth and size anomalies, especially when abnormalities were in the corners of the breast, a challenge stemming from the original analytical methodology. To overcome these challenges and not only reconstruct images but also perform classification tasks for our future blind study screening, we developed a multitask CNN. This enhanced model can evaluate depth, size, and health status based on varying phantom data, including samples without abnormalities. This approach aims to more effectively differentiate abnormal from healthy cases, thereby enhancing the practical diagnostic utility of the NIRscan probe.

## 2. Materials and Methods

### 2.1. Near-Infrared Scan (NIRscan) Probe for Breast Cancer Imaging

The NIRscan probe (Figure 1a) is an affordable, non-invasive, painless, and clinic-friendly device for early-stage breast cancer detection. It can complement existing breast cancer imaging and potentially replace the Clinical Breast Exam (CBE) through palpation. Operating in the electromagnetic spectrum’s near-infrared (NIR) region, the probe exploits biological tissues’ distinct absorption and scattering properties, allowing for deeper tissue penetration and imaging [11].

As shown in Figure 1b, the NIRscan probe features a 2048-pixel linear CCD detector and two encapsulated LEDs, providing symmetrical illumination at 690, 750, 800, and 850 nm [11]. These specific wavelengths were selected based on the large difference between extinction coefficients in the four primary components in breast tissue, including oxyhemoglobin (HbO_2_), deoxyhemoglobin (Hb), fat, and water (H_2_O) [13]. While the selected wavelengths do not perfectly align with the absorption peaks of all chromophores, they provide valuable absorption data for these tissue components. 690 nm corresponds to deoxyhemoglobin absorption, 850 nm targets oxyhemoglobin, while 750 nm and 800 nm provide information on water and fat absorption in the near-infrared range [12,14].

In addition to breast tissue imaging, the NIRscan probe’s wavelengths are useful for assessing other tissues such as bone density, blood vessels, and spinal cord soft tissue. The 800 nm wavelength can detect changes in tissue hydration, helping to assess bone density and monitor conditions like osteoporosis [15]. The probe uses 690 nm for deoxyhemoglobin and 850 nm for oxyhemoglobin to measure blood oxygen levels, aiding in detecting vascular insufficiency [15]. For spinal cord injuries, 690 nm and 850 nm wavelengths track tissue oxygenation, while 800 nm helps monitor hydration, providing insights into inflammation and recovery [16].

### 2.2. Scattering Behavior and NIRscan Imaging

Breast tissue, being a scattering medium, causes the near-infrared (NIR) light emitted from the LEDs to scatter in various directions as it interacts with tissue components such as cells, fat, and blood vessels [17]. Figure 2 illustrates how light follows a semi-circular path through the breast tissue, representing the average photon scattering behavior. This semi-circular assumption simplifies the modeling of light paths in highly scattering media. The CCD detector collects the intensity of backscattered light (reflectance), providing insights into the tissue’s composition [11].

During imaging, the reflectance data are transmitted to a connected computer (Figure 1c). The system uses a graphical user interface (GUI) that facilitates operations such as manual light source selection and CCD integration time adjustments. The data collected from 2048 pixels in the CCD array are processed to reconstruct the tissue’s internal optical properties. The reflectance data from 16 contiguous pixels are averaged to reduce noise, resulting in 128 points of raw reflectance data, which helps to smooth the signal while preserving key information. These values are used to determine the tissue’s optical properties, which can then be converted into cross-sectional optical images [11,12].

### 2.3. Study Setting and Scanning Procedure

In this preliminary study, the NIRscan probe initially measured reflectance or light intensity from 133 breast tissue-mimicking phantoms, all with fixed depth and size. The phantom used in this study is made of Delrin^®^, which simulates the optical properties of breast fat [18,19]. As illustrated in Figure 3, a 15 mm deep hole was drilled into the phantom to insert a 5 mm iron Allen key. Iron was chosen because it strongly absorbs light across a broad spectrum, including the NIR range, providing a high-contrast target to validate the system’s detection and localization capabilities. Later, an additional dataset was created, including 70 phantom cases with varying sizes and depths of abnormalities and healthy phantoms, to assess size, depth, and health status as classifiers.

As NIR light passes through the phantom and interacts with the abnormality, its intensity decreases due to high absorption and scattering, similar to tumor-affected tissue. This allows us to assess the probe’s ability to detect reflectance variations [11]. Reflectance data from cases with varying positions of the hole and different wavelengths, with the probe moving in different millimeter increments, are collected.

An imaging algorithm [11] is used to generate a 2D image (Figure 4) by mapping the calculated absorption coefficients (μa) across the tissue’s depth and length, aiding in the precise identification of anomalies such as tumors. In the 2D image, varying colors indicate different absorption levels, with high absorption (red) highlighting abnormalities. This imaging process, although detailed, is time-consuming as it involves two steps: converting reflectance data to optical properties and then translating these properties into images. To address this inefficiency, we aim to train a machine learning model to generate real-time images directly from raw reflectance data, bypassing the intermediate steps and enabling immediate diagnostic visualization.

### 2.4. Machine Learning Models for Image Reconstruction

Conventional methods for image reconstruction in Diffuse Optical Tomography (DOT) typically rely on analytical models such as the diffusion equation to estimate the optical properties of tissues. While grounded in mathematical formulations, these traditional approaches often struggle with speed and computational efficiency, making them unsuitable for real-time applications [11].

Recently, machine learning has shown considerable promise in enhancing image reconstruction accuracy, offering significant improvements over traditional methods [20]. Machine learning models can reduce computational time while maintaining high accuracy, making them more suitable for real-time imaging in medical diagnostics. Our recent study developed a polynomial regression with cross-validation to predict breast tissue absorption coefficients [11], but it could not generate accurate 3D images. In [1], an ensemble learning method was employed to identify healthy and unhealthy types, but this method failed to achieve 3D reconstruction. Feng et al. [21] developed a multilayer perceptron (MLP) feedforward neural network for 2D-DOT image reconstruction, though its performance was significantly decreased in limited-angle acquisition scenarios. Yoo et al. [22] created a CNN model to address 3D-DOT inverse scattering challenges, focusing on the nonlinearity of the problem. These advancements highlight the increasing potential of deep learning to enhance the speed and accuracy of image reconstruction, particularly in real-time clinical settings.

### 2.5. Customized CNN Model for Breast Cancer Image Reconstruction

This study develops a customized convolutional neural network (CNN) model for real-time breast cancer reconstruction using NIRscan data. A key challenge in training CNN models is the limited availability of labeled data. To address this, we implemented a structured training pipeline that includes data preprocessing, augmentation, model design, and performance evaluation to ensure robustness and generalizability.

#### 2.5.1. Data Augmentation

Initial augmentation techniques, such as rotation and adding noise, aimed to diversify the dataset. However, these transformations distort the data in ways inconsistent with the NIRscan device’s specific shape and orientation. To address this, we focused on the mirroring technique, which involves horizontally flipping images and their corresponding reflectance data. This technique preserved data integrity while introducing variations, doubling the dataset size from 133 to 266 samples.

#### 2.5.2. Data Splitting

The dataset was divided into training, validation, and test sets (see Figure 5). The training set (80% of the data) was used to fit the model, allowing it to learn patterns and adjust its weights. The validation set (10% of the data) acted as a checkpoint during training, providing unbiased feedback on the model’s ability to generalize. Finally, the test set (10% of the data) was reserved for the final evaluation to determine how well the model could perform on unseen, real-world data.

#### 2.5.3. Model Architecture and Training Process

The architecture of our custom CNN, detailed in Figure 6, showcases a comprehensive model designed for high-accuracy breast cancer image reconstruction using NIRscan data. It follows a modular approach with input preprocessing, feature extraction, refinement, and output generation stages to streamline the reconstruction task.

**Input and Feature Extraction Stages:** The model begins at the input stage, receiving reflectance data from two LED sources, each providing 128 reflectance values essential for precise image reconstruction. Each data stream is independently processed through several layers:**Conv1D layers:** Process the input reflectance data from each LED source. Conv1D layers are adept at extracting local feature patterns from sequential data, making them ideal for analyzing nuanced changes in reflectance.**Flatten layers:** Transform the structured output from Conv1D layers into a flat, one-dimensional vector.**Dense layers:** Project the flattened vectors into a higher-dimensional feature space, enabling the model to capture more complex patterns.**Reshape layers:** Reformat the high-dimensional data outputs into structured 3D outputs (128 × 128 × 3), preparing the data for the output stage.

**Output Stage (Refinement and Reconstruction):** This stage employs neural network layers to refine and enhance the extracted features:**Residual Attention Blocks:** Dynamically adjust feature importance, enhancing performance in reconstruction.**Conv2D and MaxPooling Layers:** Capture spatial relationships and reduce data dimensionality, helping avoid overfitting.**Squeeze and Excite Blocks:** Perform channel-wise recalibration of feature maps, boosting significant features.**Fusion Layer with Gating:** Integrate features with a gating mechanism to emphasize critical information for final reconstruction.

The final output is a reconstructed image of dimensions 128 × 128 × 3, generated from input reflectance data.

**Integration and Performance:** Features from both LEDs are combined into a single tensor, enabling the model to analyze patterns across the entire dataset. This integration is key to achieving high reconstruction accuracy.

#### 2.5.4. Training Process of the Custom CNN Model

The training process (Figure 7) involves several key stages:**Input Stage:** Collecting reflectance data from two LEDs.**Processing Stage:** Generates predicted images based on current model weights. Predictions are compared to ground truth using MSE and MAE.**Optimization Stage:** Backpropagation updates weights based on loss. Training typically converges around epoch 1776:**Early Stopping:** Stops training when validation loss shows no improvement for 1000 epochs.**Learning Rate Scheduler:** Reduces the learning rate by 50% after 500 epochs of stagnation (min rate = 1 × 10^−6^).

Validation loss is monitored throughout to detect overfitting and ensure effective generalization.

To handle the computational demands of this model, we utilized the NVIDIA A100 GPU, which is based on NVIDIA’s Ampere architecture. This hardware accelerator is selected for its exceptional performance capabilities tailored for high-performance computing (HPC) and deep learning workloads. The A100 GPU offers significant speed, efficiency, and scalability advantages, making it particularly well suited for the intensive parallel computations required in our CNN model’s training.

#### 2.5.5. Hyperparameter Tuning

Hyperparameter tuning begins with setting initial values for critical parameters:**Learning Rate:** Initially set to 0.0001.**Batch Size:** Configured at 8 to balance the computational load and training dynamics.**Epochs:** The model is set to train for up to 2000 epochs as the upper limit, providing sufficient time for learning and adjustments. However, it converged by epoch 1776 due to early stopping, triggered when no improvement in validation loss was observed after a patience period of 500 epochs.

The dataset is systematically split into training, validation, and test sets to evaluate the model’s performance comprehensively:**Training and Validation:** During these phases, hyperparameters are adjusted based on the performance metrics observed from the validation dataset.**Testing**: Once training is finalized, the model’s performance is evaluated on the test set using various metrics to assess accuracy and generalization capabilities.

#### 2.5.6. Evaluation Metrics

The following metrics are systematically employed to provide a comprehensive assessment of the model’s performance, reflecting its efficacy and readiness for real-world applications:**Total Inference Time:** Measures the complete duration required to process the test set, reflecting the model’s efficiency.**Frames Per Second (FPS):** Indicates how many frames the model can process per second, which is important for real-time analysis applications.**Time Per Frame:** The time taken to process a single frame is important for evaluating the model’s performance in real-time settings.**RMSE (Root Mean Square Error):** Quantifies the average magnitude of the prediction error, providing a sense of how far off the predictions are from actual values.**MAE (Mean Absolute Error):** Measures the average magnitude of prediction errors, representing how close the predictions are to the actual outcomes without considering direction.**R^2^ (Coefficient of Determination):** Indicates the proportion of variance in the dependent variable predictable from the independent variables, providing insight into the goodness of fit.**Fuzzy Jaccard Index:** Compares the predicted and actual values to quantify the similarity and diversity, assessing the accuracy of predictions.

The outcomes from these evaluations will be detailed in the results section, providing valuable insights into the model’s real-world applicability and performance.

### 2.6. Multitask CNN Model for Breast Cancer Image Reconstruction and Classification

Our initial custom CNN aimed to predict real-time images but encountered challenges in accurately estimating tumor depth and size, mainly when abnormalities were located at the breast’s corners. These issues stemmed from limitations in the original analytical model, which assumed a semi-circular path for light propagation. This assumption introduced geometric distortion, resulting in depth misestimations, especially in peripheral regions.

To address this, we enhanced the dataset with phantom experiments using Allen keys of varying diameters and depths, enabling the creation of more accurate depth and size categories. We also incorporated data from healthy phantoms without abnormalities to broaden the model’s training scope and improve overall dimensional accuracy.

Building on our previous work, this section introduces a significant enhancement to our custom CNN model, expanding its capabilities to reconstruct breast cancer images and classify them based on depth, length, and health status. This model uses shared layers and multiple output heads to perform tasks simultaneously.

**Model Enhancements and Integration:** The architecture of the updated model now includes multiple output layers, each designed to handle a specific classification task along with image reconstruction. This integration equips the model to provide comprehensive diagnostic information, which is crucial for clinical accuracy. Specifically, the model classifies the following:**Length Classification**: Measures the length of abnormalities, categorizing them into 0 mm for healthy cases and 1.5 mm, 3 mm, and 5 mm (Figure 8) for different diameters of unhealthy tissue abnormalities (Allen keys).

**Depth Classification**: The use of a semi-circular model in the original analytical model for training the CNN can lead to depth distortions, especially because this ‘banana shape’ effect causes errors in depth when the probe is positioned at the corners of the phantom. As shown in Figure 9a, although the actual depth of an abnormality might be 15 mm, corner positioning can cause the red region, which represents the abnormality, to skew. This results in a perceived shallower depth (Figure 9b). To address this issue, we implemented a model that utilizes actual depth measurements for training, which enables the prediction of correct depths despite the distorted appearances in the image reconstructions. Furthermore, our classifier incorporated additional depth layers to accommodate various depths and enhance the model’s accuracy. These layers are trained on specific depth measurements to ensure accurate depth predictions.

**Health Status Classification**: A binary classification distinguishes between healthy and unhealthy phantoms. Abnormalities are shown as dark red regions (see Figure 8). In contrast, the absence of abnormalities appears as solid blue when corner artifacts from non-collimated light are removed (see Figure 10a for artifacts and Figure 10b for artifact removal). In our experiments, we minimized low noise and clarified images, though some training images retained minor noise. In both scenarios, the healthy tissue is visible.

These classification tasks are treated with the same importance as the reconstruction of breast cancer images. The training process now incorporates a multi-output strategy, ensuring that each task receives appropriate attention during the learning phase. Shared layers allow for feature extraction that benefits all tasks, enhancing the generalization across different data aspects.

**Evaluation Metrics for Multitask CNN Model:** We used a comprehensive set of evaluation metrics, including Inference Metrics and Image Reconstruction Metrics, similar to those used in our customized CNN. Additionally, we introduced classification metrics tailored to the specific tasks of the model: depth and length classifications are evaluated using accuracy to assess how often the model predicts the correct category, reflecting its efficiency in measuring physical dimensions. For the binary classification of health status—distinguishing between healthy and unhealthy tissues—accuracy, precision, recall, and the F1 Score are used. Detailed results of these evaluations are presented in the subsequent section.

## 3. Results

This section presents the evaluation results of our CNN models. Section 3.1 details the performance of the custom CNN on 266 phantom cases, while Section 3.2 presents the results of the multitask CNN on 70 phantom cases with varying sizes, depths, and classification statuses. In all figures, the vertical axis represents tissue depth (mm), and the horizontal axis represents tissue length (mm). However, the axes are not shown in the result images, as they were removed during training to match the input format of the machine learning model and eliminate non-informative visual features.

### 3.1. Custom CNN

Table 1 presents the performance metrics of the custom CNN, highlighting its effectiveness in terms of inference time, frame rate, and accuracy. Additionally, Figure 11 compares the ground truth with the corresponding CNN-generated images for the phantom dataset.

### 3.2. Multitask CNN

The performance metrics for the 70 phantoms, which vary in depth, length, and health status, are presented in Table 2. This table highlights the model’s effectiveness across various metrics, including inference times, image reconstruction metrics, and classification accuracies for depth, length, and health status. We observe perfect classification accuracy for depth and health status, but only 92.86% for diameter. Compared to the custom CNN, although the inference times are faster due to the smaller number of images, the error rate is slightly higher due to the limited amount of training data.

Additionally, Table 3 visually compares the ground truth with the reconstructed images at a constant depth, illustrating two cases, with and without an Allen key, representing unhealthy and healthy cases. Corner artifacts exist in both healthy and unhealthy cases. However, when an abnormality is present, the high-intensity (red color region) significantly surpasses corner noise, rendering the corner artifacts less noticeable. Conversely, corner noises remain visible in healthy cases lacking such an abnormal high intensity.

We addressed distortions in diameter and depth within corner images by training with precise measurements. For instance, Table 4 illustrates cases with depth distortions, where the original depth is 15 mm. We applied this approach to two Allen key cases with predicted diameters of 3 mm and 1.5 mm. The multitask CNN effectively predicts abnormalities in diameter and depth, ensuring that the measurements accurately reflect the actual dimensions, even in the presence of original corner distortions. Consequently, the diameter classification accuracy achieved 92.86%, with only one case incorrectly predicted, while the model attained perfect classification for depth.

## 4. Discussion

In this study, we utilized solid phantoms made of Delrin to simulate the optical properties of adipose breast tissue, with a primary focus on detecting and localizing abnormalities (represented by an embedded Allen key) and evaluating the performance of our real-time imaging system. Although our research also extends to liquid phantoms to simulate a range of breast tissue densities—from adipose to fibroglandular tissue—and includes patient-based data, these results are not presented in this paper. This exclusion ensures a focused discussion of the solid phantom-based evaluation, aligning with the study’s scope.

The custom CNN model demonstrated robustness and effectiveness on the solid phantom dataset, showcasing strong inference performance with lower RMSE and higher R^2^ values. These results demonstrate precise image reconstruction under controlled conditions, underscoring the model’s potential for clinical diagnostic applications where accuracy and reliability are paramount.

Our enhanced dataset confirms the model’s ability to adapt and learn from augmented data, illustrating the effectiveness of our data augmentation techniques. The computational efficiency of our model is highlighted by its rapid inference times, suitable for real-time clinical applications, facilitated by the NVIDIA A100 GPU’s high-performance computing capabilities.

Significant advancements were achieved with the development of a multitask CNN model, which not only reconstructs images but also classifies them based on depth, length, and health status. This model leverages the robust feature extraction capabilities of CNNs to perform multiple complex tasks simultaneously, thereby enhancing both diagnostic efficiency and accuracy.

The multitask model features multiple output layers tailored for specific classification tasks, as well as image reconstruction. We have refined the model’s classification abilities for depth and diameter using predefined categories, addressing inaccuracies introduced by the semi-circular light path assumption. Plans are underway to integrate more complex geometrical models into our machine learning framework, enhancing the accuracy of light propagation mapping across various tissue types. This is expected to improve depth and diameter predictions significantly.

Furthermore, we expanded the classification capabilities to include health status, distinguishing between healthy and abnormal tissue with a multi-output strategy that ensures balanced training across tasks and enhances generalization.

The model and its weights have been preserved, preparing it for future applications on unseen datasets. This readiness supports its use in prospective studies, potentially including blind trials where individuals’ health status is unknown, thus enhancing its utility for broader screening and diagnostic applications.

In summary, the earlier versions of the NIRscan probe faced challenges in real-time tumor localization and required manual adjustments to offset artifacts. Our current model overcomes these hurdles by incorporating advanced computational strategies and hardware enhancements, which have significantly improved imaging speed and accuracy. This approach aims to more effectively differentiate abnormal cases from healthy cases, thereby enhancing the practical diagnostic utility of the NIRscan probe.

## 5. Conclusions

This study demonstrates the effectiveness of a customized convolutional neural network (CNN) model, integrated with NIRscan technology, for real-time breast cancer detection. The CNN model was tested on solid phantom data and exhibited strong performance metrics: an RMSE of 0.0624, MAE of 0.0360, R^2^ of 0.9704, and Fuzzy Jaccard Index of 0.9121. These results underscore the model’s precision and reliability for clinical applications in controlled settings.

Further, introducing a multitask CNN model significantly expanded our system’s diagnostic capabilities. This model reconstructs images and performs classifications based on depth, length, and health status. By leveraging the robust feature extraction capabilities of CNNs, the multitask model efficiently handles multiple diagnostic tasks simultaneously, which is crucial for enhancing diagnostic accuracy and operational efficiency in clinical settings.

The proven accuracy and efficiency of the multitask model underscore its potential to enhance breast cancer screening and diagnostics, moving beyond traditional imaging techniques towards more dynamic and precise real-time imaging.

While initial evaluations were based on phantom data, the promising results suggest the model’s applicability to real-world clinical scenarios. Planned future studies aim to deploy this model in patient trials to generate immediate, real-time diagnostic images, further streamlining the diagnostic process and enhancing the practical utility of the NIRscan technology for non-invasive breast cancer screening.

## Figures and Tables

**Figure 1 sensors-25-02349-f001:**
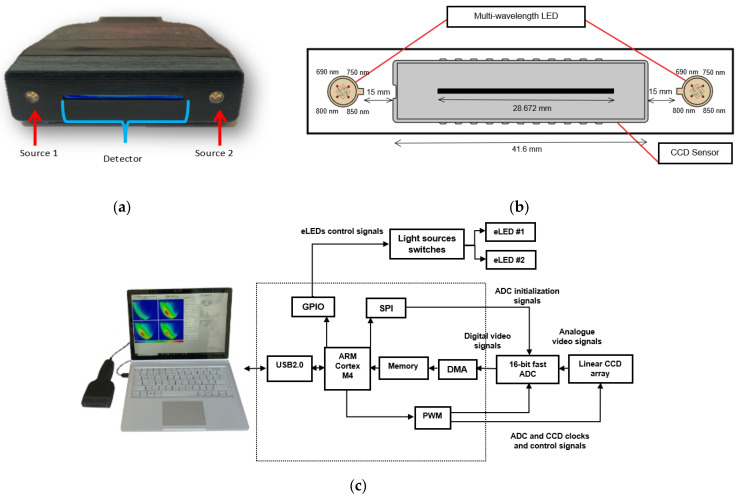
Opti-Scan probe and system overview: (**a**) front view of the probe, showing light sources and linear CCD; (**b**) diagram of the CCD sensor and LED arrangement in the probe head; (**c**) photo of the NIRscan probe connected to a laptop running custom software, with a schematic diagram of the sensory system highlighting key components [12].

**Figure 2 sensors-25-02349-f002:**
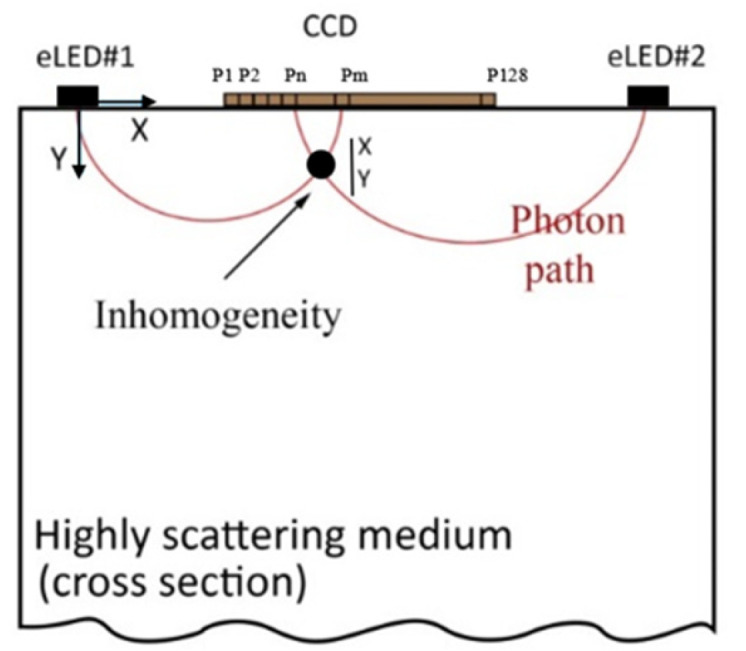
Sketch of light propagation in highly scattering tissue.

**Figure 3 sensors-25-02349-f003:**
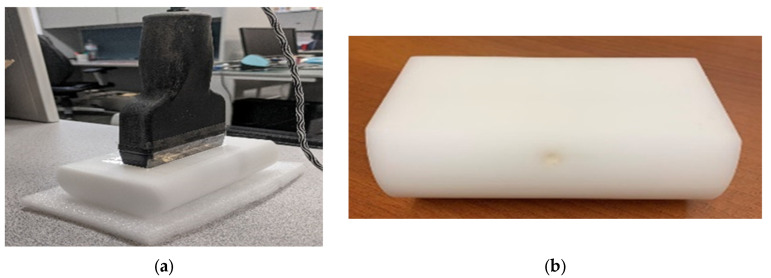
(**a**) The probe collects data from the Delrin phantom. (**b**) Delrin has a hole for inserting a key.

**Figure 4 sensors-25-02349-f004:**
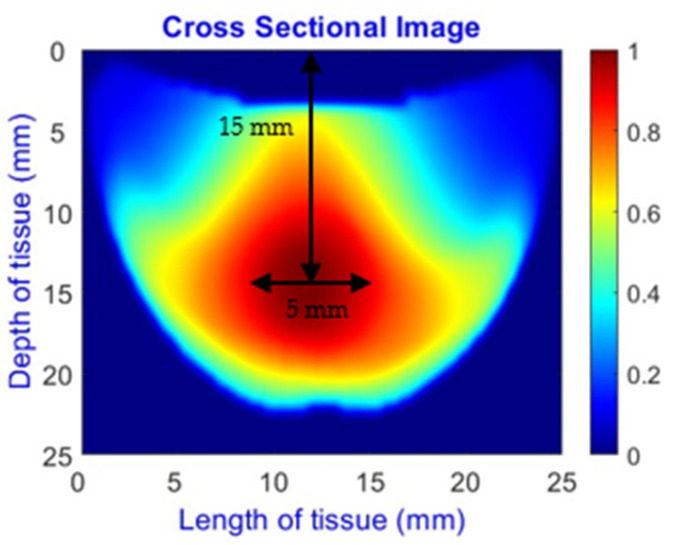
Absorption distributions showing a 5 mm Allen key at a depth of 15 mm.

**Figure 5 sensors-25-02349-f005:**
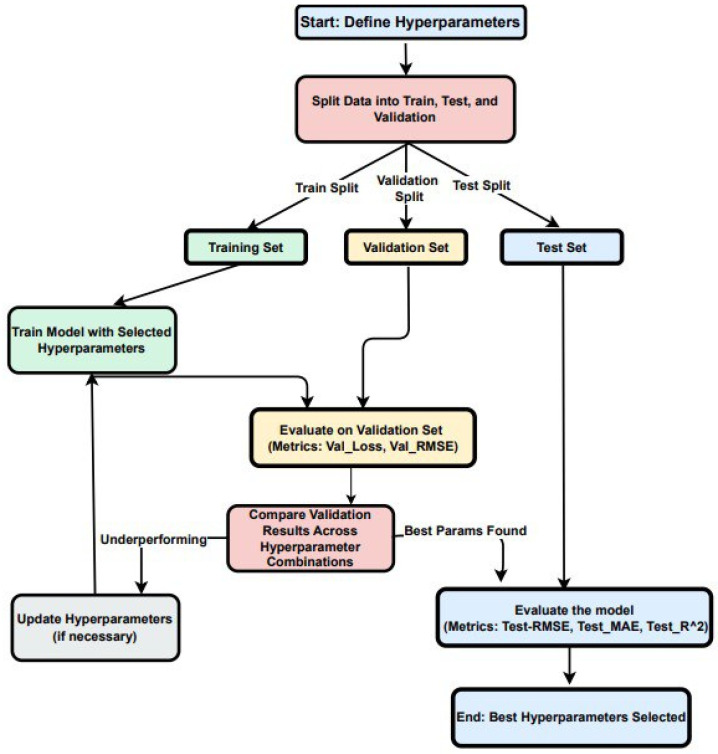
Data split, hyperparameter tuning, and evaluation workflow for the custom CNN.

**Figure 6 sensors-25-02349-f006:**
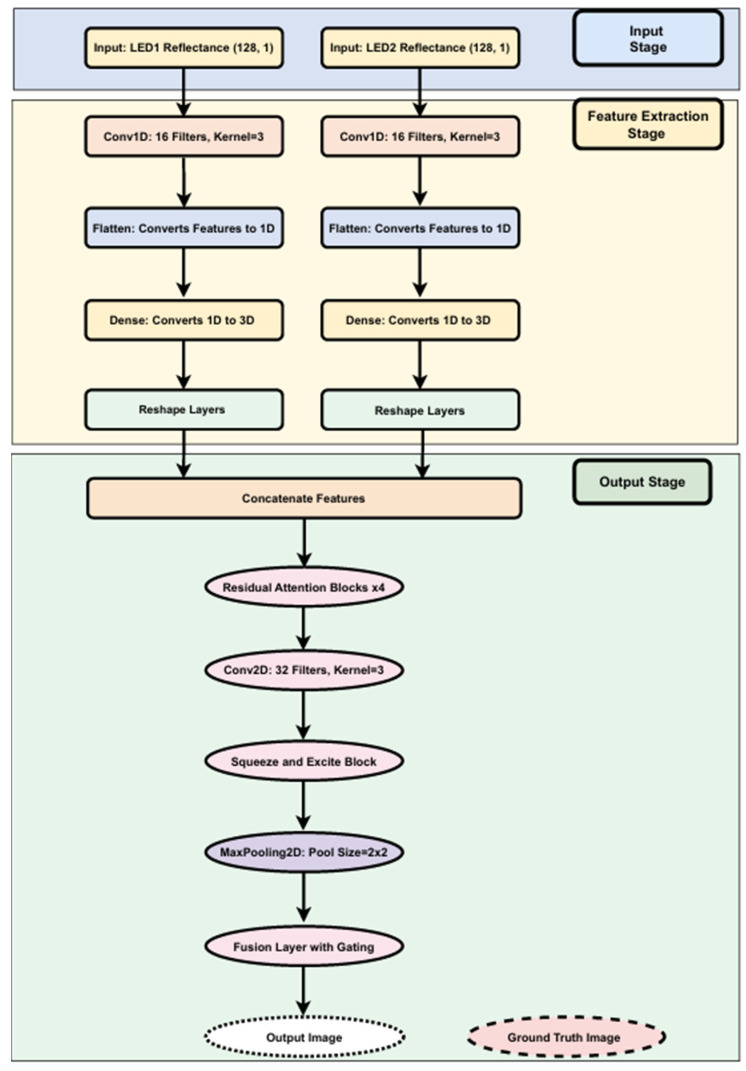
Structural overview of the custom CNN model with residual attention and fusion.

**Figure 7 sensors-25-02349-f007:**
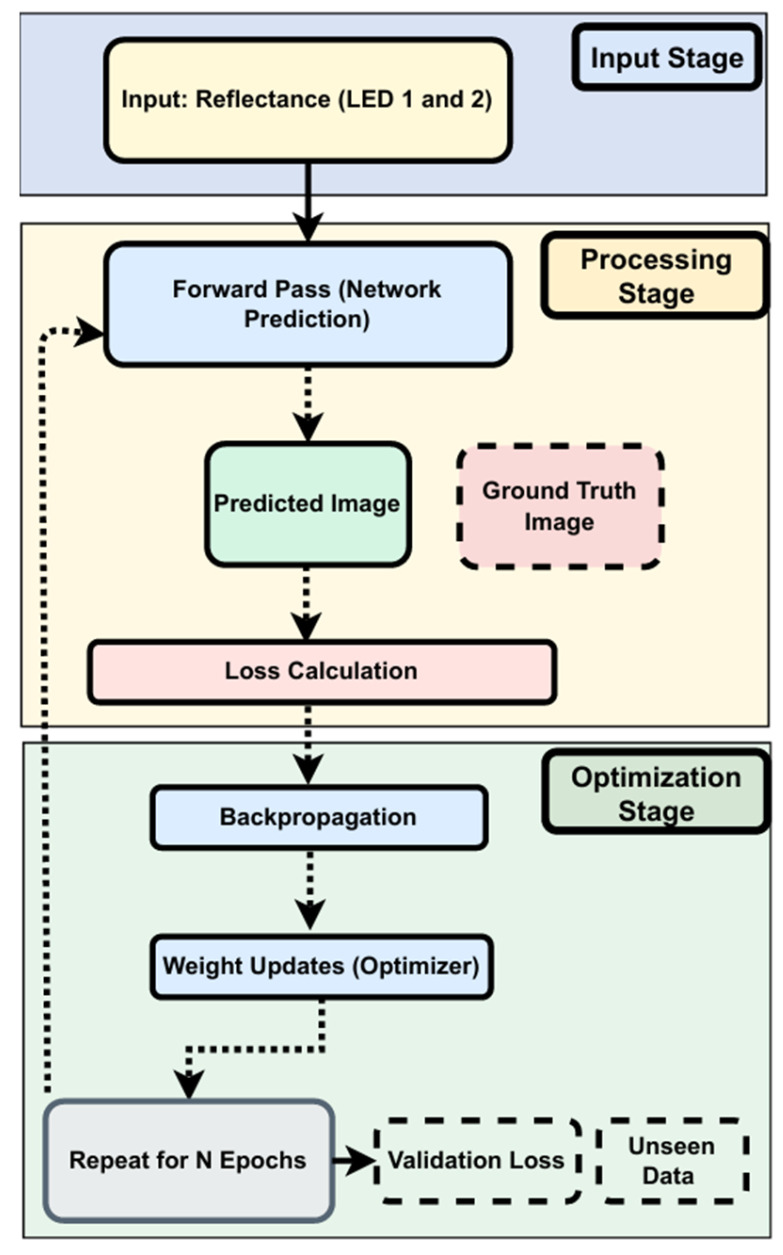
Training process of the custom CNN.

**Figure 8 sensors-25-02349-f008:**
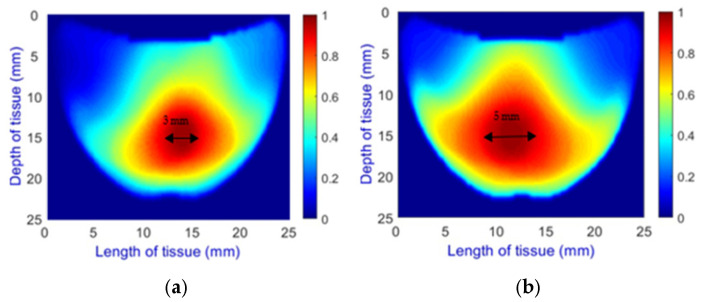
Comparison of Allen key diameters of (**a**) 3 mm and (**b**) 5 mm inside the phantom.

**Figure 9 sensors-25-02349-f009:**
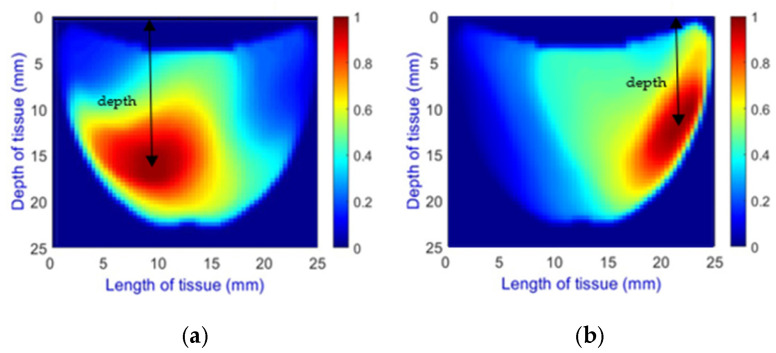
Depth distortion at corner positions: (**a**) Original depth (15 mm) and (**b**) Distorted depth.

**Figure 10 sensors-25-02349-f010:**
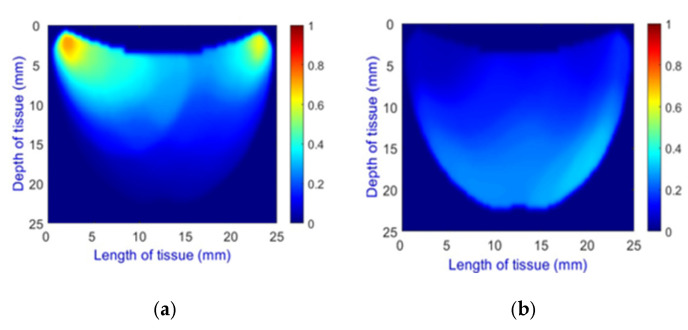
A phantom without any abnormality: (**a**) With corner artifacts (**b**) After artifact removal.

**Figure 11 sensors-25-02349-f011:**
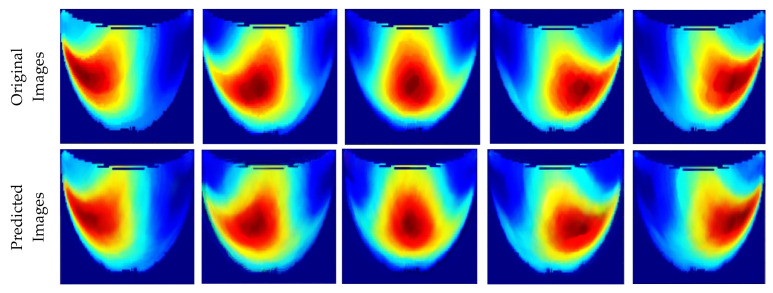
Comparison between the original and predicted images generated by the custom CNN.

**Table 1 sensors-25-02349-t001:** Performance Metrics of the Custom CNN.

Metric Description	Custom CNN
Total Inference Time	1.6229 s
Frames Per Second (FPS)	63.60
Time Per Frame	0.0157 s
Average RMSE	0.0624
Average MAE	0.0360
Average R^2^	0.9704
Fuzzy Jaccard Index	0.9121

**Table 2 sensors-25-02349-t002:** Performance metrics of the multitask CNN across varying depth, length, and health status classifications.

Metric Description	Multitask CNN
Total Inference Time	0.0886 s
Frames Per Second (FPS)	157.94
Time Per Frame	0.0063 s
Average RMSE	0.1552
Average MAE	0.0831
Average R^2^	0.8244
Fuzzy Jaccard Index	0.8114
Depth Classification Accuracy	1.0000
Diameter Classification Accuracy	0.9286
Health Classification Accuracy	1.0000
Health Classification Precision	1.0000
Health Classification Recall	1.0000
Health Classification F1 Score	1.0000

**Table 3 sensors-25-02349-t003:** Comparison of the original and predicted images with varying health status.

Original Image	Predicted Image	Original Measurements	Predicted Measurements
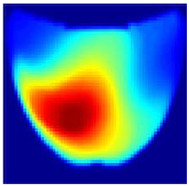	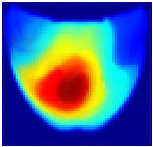	Depth: 15 mm Diameter: 3.0 mm Status: Unhealthy	Depth: 15 mm Diameter: 3.0 mm Status: Unhealthy
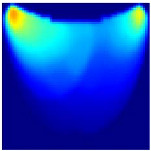	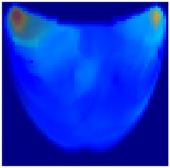	Depth: 15 mm Diameter: 0.0 Status: Healthy	Depth: 15 mm Diameter: 0.0 Status: Healthy

**Table 4 sensors-25-02349-t004:** The abnormality’s diameter and depth appear distorted at the corner, but the classifier correctly identified the measurements.

Original Image	Predicted Image	Original Measurements	Predicted Measurements
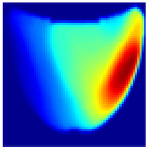	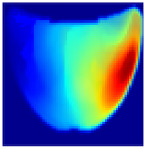	Depth: 15 mm Diameter: 3.0 mm Status: Unhealthy	Depth: 15 mm Diameter: 3.0 mm Status: Unhealthy
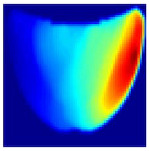	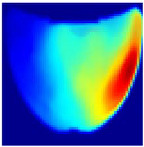	Depth: 15 mm Diameter: 1.5 mm Status: Unhealthy	Depth: 15 mm Diameter: 1.5 mm Status: Unhealthy

## Data Availability

Data are contained within the article.

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
