# Peer review of "A Multitask CNN for Near-Infrared Probe: Enhanced Real-Time Breast Cancer Imaging"

_sensors, 2025, doi:10.3390/s25082349_

Round 1
Reviewer 1 Report
Comments and Suggestions for Authors
This article focusses on the use of a near-infrared imaging sensor and an advanced convolutional neural network model to improve the precision of breast cancer diagnosis, especially in cases with dense tissues. The study proposes a CNN multitask approach that simultaneously reconstructs images and classifies tumours based on depth, length, and health status. There are a few comments below.
1. Line 49: In the literature review, it is reasonable to also consider the microwave radiometry method, which allows one to obtain data on the internal temperature of biological tissues.
2. Line 138: Using a semi-circular model to describe the path of light through breast tissue may be an oversimplification. It is worth considering whether this model is adequate in the case of complex breast structures, such as those with different tissue densities or heterogeneous areas. What is the error that arises from this simplification and how can it affect the accuracy of the reconstruction?
3. Line 146: Whether the use of specific wavelengths affects the uncertainty of the measurement. Possible errors or influences that may arise due to changes in external factors, such as ambient temperature, tissue density, or individual differences between patients, should be mentioned.
4. Line 163: The study used phantoms made by Delrin to mimic the optical properties of adipose breast tissue. How adequate is the use of Delrin to simulate different types of breast tissue, such as fibrous or dense tissue? Have comparisons been made with other materials that more accurately simulate different types of breast tissue?
5. Line 165: Iron has been used to simulate tumours because it strongly absorbs light over a wide spectrum. Is it important to consider that iron has specific optical properties that are different from those of human tissues such as blood vessels or tumour masses? How might this affect diagnostic accuracy, especially in a real-world clinical trial setting?
6. Line 365: Problems with the accuracy of predicting tumour depth and size occurred, particularly when abnormalities were located at the corners of the breast. What specific limitations in the original analytical methodology led to these problems? Has additional research been conducted to better understand how the angle or location of the abnormality affects the accuracy of the prediction?
7. In the 'Discussion' section, you should add a comparison of the results obtained in the article with the results of other studies on this topic.
Reviewer 2 Report
Comments and Suggestions for Authors
General Comments
The authors have made a bold statement in the final sentence of their abstract, concluding that their "results underscore the potential of NIRscan technology coupled with a sophisticated multitask CNN to revolutionize real-time breast cancer screening and diagnostics." However, having read the full paper that follows, I cannot agree with this statement. I was certainly not convinced that what they have developed will be a game changer in the screening for breast cancer. Here are some of my concerns.
First, although the authors have included 39 patients in their study, there was absolutely no attempt made to compare their NIRscan technique with the traditional imaging techniques such as mammography or ultrasound. Why was this the case? Surely, all the women would have had a mammogram at the very least?
Second, it is extremely difficult to envisage exactly how the NIRscan technique is applied to a human patient. The one image that supposedly shows this is Figure 5 where, as far as I can tell, the probe is applied to a phantom breast that is part of a dummy. It's not clear whether the probe is simply held in a stationary position or is it scanned in a direction that's orthogonal to the charge coupled device (CCD)?
Third, one of the well-known limitations when using infrared imaging is the limited depth penetration. There is no mention of this in the paper except perhaps in Figure 6 which would suggest that the NIRscan device can measure a depth of 25 mm. If this is indeed the limit, then this penetration depth is insufficient if the lesion is located close to the chest wall at a depth well in excess of 25 mm.
Fourth, the authors state (lines 54 to 56) that their NIRscan technology is protected by US Patent US20220409058, also identified as application 17/872,299. I did some checking on the website of the United States Patent & Trademark Office (USPTO) -- since I have my own Customer Number -- and I discovered that this application was issued with a Final Rejection by the examiner on 15 April 2024 and then Abandoned on 20 December 2024 (see attachment). It would therefore appear that the authors do not have the protection they are claiming and this reference is of dubious value.
Fifth, the font employed in many of the figures is far too small, and in some cases the text is simply illegible. I refer here to Figures 1(d), 7, 8, and 9. In addition, Figure 6 -- which has the headline "Cross Sectional Image" -- has two axes marked Depth of tissue (mm) and Length of tissue (mm), it's not clear whether Figures 10, 11, 12, 13, 14, 15 and 16 have the same axes. It's also unclear which Length is being referred to, particularly with respect to anatomical axes.
Sixth, I was extremely puzzled why the phantom data were combined with the patient data. Surely the patient data should have been studied -- and presented -- separately, and as stated above, it would have been reassuring to see the patient data from the NIRscan system compared with image data from either mammography or ultrasound.
Finally, I am concerned that the references are riddled with errors, mostly of omission, which strikes me as extremely unprofessional. Further details are provided below.
Specific Comments
line 39 ... may be missed, leading to false ...
line 99 ... predicting depth and size anomalies, especially ...
Figure 8 Structural Overview of the custom CNN model ...
Figure 14 It's not clear which images are of phantoms and which are of patients
line 596 At which university was the PhD by Shokoufi earned?
line 604 What date was the online article by Mayo published?
line 617 What are the page numbers for the paper by Momtahen?
line 622 At which university was the PhD by Momtahen earned?
line 625 This reference appears to be incomplete (1950008?)
line 626 Where was the paper by Zhao published?
line 632 Where was the paper by Jee published?
line 635 What date was the online article "Shining a ..." published?
line 644 The article by Lim has no source and no date!

Round 2
Reviewer 1 Report
Comments and Suggestions for Authors
Accept in present form.
Reviewer 2 Report
Comments and Suggestions for Authors
I have carefully read the authors' answers to all my concerns and I am satisfied with their response.